# A systematic review of whole disease models for informing healthcare resource allocation decisions

**Huajie Jin**[1]*, **Paul Tappenden**[2], **Xiaoxiao Ling**[3], **Stewart Robinson**[4], **Sarah Byford**[1]

**1** King's Health Economics (KHE), Institute of Psychiatry, Psychology & Neuroscience at King's College London, London, United Kingdom, **2** Health Economics and Decision Science, School of Health and Related Research, University of Sheffield, Sheffield, United Kingdom, **3** Department of Statistical Science, University College London, London, United Kingdom, **4** Newcastle University Business School, Newcastle, United Kingdom

* huajie.jin@kcl.ac.uk

**Data Availability Statement:** All relevant data are within the manuscript and its Supporting Information files.

**Funding:** The authors received no specific funding for this work. XL receives PhD scholarship (EP/

## Abstract

### Background

Whole disease models (WDM) are large-scale, system-level models which can evaluate multiple decision questions across an entire care pathway. Whilst this type of model can offer several advantages as a platform for undertaking economic analyses, the availability and quality of existing WDMs is unknown.

### Objectives

This systematic review aimed to identify existing WDMs to explore which disease areas they cover, to critically assess the quality of these models and provide recommendations for future research.

### Methods

An electronic search was performed on multiple databases (MEDLINE, EMBASE, the NHS Economic Evaluation Database and the Health Technology Assessment database) on 23rd July 2023. Two independent reviewers selected studies for inclusion. Study quality was assessed using the National Institute for Health and Care Excellence (NICE) appraisal checklist for economic evaluations. Model characteristics were descriptively summarised.

### Results

Forty-four WDMs were identified, of which thirty-two were developed after 2010. The main disease areas covered by existing WDMs are heart disease, cancer, acquired immune deficiency syndrome and metabolic disease. The quality of included WDMs is generally low. Common limitations included failure to consider the harms and costs of adverse events (AEs) of interventions, lack of probabilistic sensitivity analysis (PSA) and poor reporting.

R513143/1) from the Engineering and Physical
Sciences Research Council (EPSRC), UK (https://
www.ukri.org/councils/epsrc/). EPSRC had no role
in study design, data collection and analysis,
decision to publish, or preparation of the
manuscript. The authors have no competing
interests to declare that are relevant to the content
of this article.

**Competing interests:** The authors have declared
that no competing interests exist.

## Conclusions

There has been an increase in the number of WDMs since 2010. However, their quality is
generally low which means they may require significant modification before they could be
re-used, such as modelling AEs of interventions and incorporation of PSA. Sufficient details
of the WDMs need to be reported to allow future reuse/adaptation.

## Introduction

Health economic models are routinely used to inform healthcare resource allocation decisions
in many countries across the world [1–5]. Models provide an explicit means of structuring a
decision problem and synthesising all relevant evidence to estimate the expected costs and
consequences of alternative health care interventions within a given health condition, usually
over a lifetime horizon. Conventional health economic models are 'piecewise' in that they typi-
cally address a single decision problem at a specific decision point in a care pathway. 'Piece-
wise' models represent the standard analytic approach for informing decisions about the
availability of health technologies by the National Institute for Health and Care Excellence
(NICE) and similar agencies elsewhere [6–8], but they are subject to several limitations [9].
The first of these relates to the failure to capture system interdependencies between different
interventions. The cost-effectiveness of any new intervention is dependent not only on the
costs and effectiveness of the new intervention itself, but also on the configuration of the pre-
vailing system, i.e. the availability, costs and effectiveness of existing interventions [9, 10]. For
example, the cost-effectiveness of a new test for a given cancer type may be dependent on cur-
rently recommended treatment options for patients with diagnosed disease, as well as the
availability of a screening programme for asymptomatic individuals. This type of system-level
interdependency between interventions used for the same condition is seldom adequately cap-
tured by piecewise models due to their limited scope. Second, piecewise models often employ
a simple piecewise cost per quality-adjusted life year (QALY) threshold rule which does not
explicitly consider the budget constraint [11, 12]. However, it has been well documented that
the repeated application of a threshold-based decision rule may lead to uncontrolled growth in
health-care expenditure [13–18]. Third, most models are developed with the intention of
informing a single decision problem within a broader pathway of care. This means that across
a whole disease area, reimbursement and coverage decisions are based on a number of asyn-
chronously developed discrete economic models which tend to apply different model struc-
tures, assumptions and evidence. This can lead to a situation whereby two models addressing
the same decision question produce inconsistent conclusions, with potential to lead to sub-
optimal adoption decisions [19–25].

System-level models, which include important events, health outcomes and costs across an
entire disease area, represent a potential means of addressing the limitations of conventional
piecewise models. Three well-known examples of system-level models include the US Archi-
medes diabetes model [26], the US Coronary Heart Disease (CHD) Policy Model [27], and the
UK CHD model [28]. Although this type of modelling approach dates back to 1977 [29], it was
not well-defined until 2012 when Tappenden *et al.* set out a methodological framework for
whole disease models (WDM) [9]. In short, a WDM is a system-level generic disease model
which allows for the consistent economic analysis of options across entire disease and treat-
ment pathways, including prevention, detection, diagnosis and treatment [9]. Owing to the
broader scope of these models, which focus on the whole disease and treatment pathway rather

than individual decisions within that pathway, WDMs can provide a consistent conceptual and mathematical platform for the economic analysis of a large number of health care interventions based on a single model. In addition, WDMs can allow for the consideration of a variety of different economic decision rules which jointly deal with investment and disinvestment decisions through reference to a budget constraint, such as the disease-level constrained maximisation of health decision rule [30, 31]. Under this decision rule, different combinations of healthcare services (investment and disinvestment options) are tested [9]. The one that maximises health benefits while staying within the available budget is considered the most cost-effective choice.

Despite the potential benefits of the approach, the development of a WDM requires a significant initial investment of time and resources [9, 32] and presents additional challenges for model verification and validation. The initial investment in WDMs is therefore of greater value if they are re-used and adapted over time. However, re-use requires other modellers to be aware of existing WDMs and to determine their quality, as this will impact on whether the WDMs can be re-used. To our knowledge, no previous studies have systematically reviewed which WDMs exist, or the extent to which previously developed models could be re-used to address future decisions. To fill this gap, this systematic review aims to identify existing models meeting the criteria of a WDM for any disease (regardless of whether they were labelled as a WDM), to critically examine the quality of these models and to provide recommendations to improve the quality, reporting, and adaptability for future WDMs.

## Methods

This systematic review was conducted according to the PRISMA recommendations for reporting systematic reviews and meta-analyses of studies that evaluate healthcare interventions [33]. The protocol of this review was registered with PROSPERO (https://www.crd.york.ac.uk/prospero/) on 12[th] October 2020 (CRD42020199875).

### Inclusion/exclusion criteria

Inclusion and exclusion criteria were defined *a priori*. It was hypothesised that models which strictly meet the criteria of a WDM defined by Tappenden *et al* [9] may be rare. Therefore, for the purpose of this systematic review, we decided to include studies which report models that broadly meet the criteria of being a WDM, i.e. a model which can evaluate multiple decision points (i.e. ≥3) for people with a given disease, and people at risk who may, or may not, go on to develop the health condition, and thus evaluates both the prevention and treatment of the condition simultaneously. Throughout this paper, these are referred to as 'WDMs', regardless of whether they were reported as a WDM in the original study. Those narrower models which evaluate multiple decision points (i.e., ≥3) only for people with a given disease (thus excluding people at risk of the disease), or only for people at risk of the disease (and thus excluding people with the given disease), are referred to as 'pathways models'. These pathways models were excluded from the review as they do not meet our inclusion criteria of a WDM; however, a brief summary of identified pathways models is provided.

A model was assessed as meeting the criteria of a WDM either by demonstration (i.e., the authors used the model to address three or more decision points and reported the results in the paper) or based on authors' reporting (i.e., the authors did not use the model to address evaluate options at multiple decision points in the paper, but they clearly reported that the model can be used in such a way). No limits were applied to the searches or the review inclusion criteria regarding specific diseases or conditions under consideration, types of economic evaluation, population, intervention or comparator, or outcome measures. Only published

papers were included. Studies were excluded if they met any of the following criteria: (i) reviews, commentaries, letters, editorials, or abstracts; or (ii) not reported in English.

Whether a paper meets our inclusion criteria or not was determined based on the content of each individual paper, rather than the content of a series of related papers. Therefore, models which were used to address multiple decision points in a series of papers, but each paper only used that model to address one or two decision points were excluded.

### Search strategy

Electronic biomedical databases searched included MEDLINE (including in-Process & other non-indexed) and EMBASE which were searched through the Ovid interface (https://ovidsp. ovid.com/). In addition, the NHS Economic Evaluation Database (NHS EED) and the Health Technology Assessment (HTA) Database were searched, accessed through the Cochrane library interface (http://onlinelibrary.wiley.com/cochranelibrary/search8). The search strategies included Medical Subject Heading (MeSH) terms and text words. Each follows a similar structure: economic evaluation-related terms AND WDM-related terms AND limitation terms about human studies and English language. The original searches and two update searches were conducted on 21 July 2020,18th July 2022, and 23rd July 2023 respectively. No restriction by publication year was applied. The full search strategy is reported in S1 Text. The reference lists of all identified WDMs were hand searched for further relevant studies which may have been missed by the electronic searches. Retrieved search results were downloaded into Endnote reference management software (Clarivate Analytics, version X9.3.3).

### Assessment of abstracts and papers for inclusion

Screening of abstracts and papers against the inclusion criteria was undertaken independently by two reviewers (HJ and XL). Final inclusion of studies in the review was determined by agreement of both reviewers, with disagreements resolved by discussion.

### Data extraction and analysis

Data were extracted by one reviewer (HJ) and checked by a second reviewer (XL), with disagreements resolved by discussion. The following information was extracted using Microsoft Excel (Microsoft Corporation, Microsoft 365 subscription) from all included studies: author; year; country; disease area; whether the model met the WDM or pathway model criteria by demonstration or authors' reporting; number of decision points addressed; type of economic evaluation; main effectiveness outcome; modelling method; software; economic perspective; decision rule(s) used; affiliation of the corresponding author; and other information relevant for quality assessment. Study characteristics were summarised descriptively.

### Quality assessment

Four commonly used checklists for economic evaluations were considered for the quality assessment of the current review, including the BMJ checklist (or Drummond checklist) [34], the Consolidated Health Economic Evaluation Reporting Standards (CHEERS) statement [35], Philips's checklist [36], and the NICE checklist [37]. Of these four checklists, the NICE checklist [37] was deemed to be most appropriate for the current review because (i) it focuses on the methodological quality of studies, as opposed to reporting quality (e.g. the CHEERS statement [35] focuses on the reporting quality rather than the methodological quality); (ii) it is appropriate for modelling studies, as opposed to trial-based economic evaluation (e.g. the BMJ checklist [34] is more appropriate for trial-based economic evaluation); and (iii) it allows

the users to make an overall judgement regarding the methodological quality of the studies assessed. Therefore, it is easier to summarise and compare the methodological quality of a large number of included studies using the NICE checklist, compared to those checklists which do not provide an overall judgement regarding the methodological quality of the studies assessed (e.g. Philips's checklist [36]).

The NICE checklist consists of two sections: Section 1 aims to assess the applicability of a study to the decision problems that need to be addressed, whilst Section 2 aims to assess the methodological quality of the study. Given that the aim of the review was to assess the methodological quality of the included study, rather than to assess the applicability of the model results to the UK setting, only Section 2 of the NICE checklist was used. Section 2 consists of twelve quality criteria and an overall assessment. Based on the number and importance of quality criteria that a study fails, an assessment regarding the overall methodological quality of the study can be classified into one of the following categories: (i) very serious limitations–the study fails to meet one or more quality criteria, and this is highly likely to change the conclusions about cost-effectiveness; (ii) potentially serious limitations–the study fails to meet one or more quality criteria, and this could change the conclusions about cost-effectiveness; and (iii) minor limitations–the study meets all quality criteria, or fails to meet one or more quality criteria but this is unlikely to change the conclusions about cost-effectiveness.

Two reviewers (HJ and XL) performed quality assessment for all included studies, with disagreements resolved by discussion.

## Results

### Study identification and selection

The detailed results of the literature search are reported in S1 Text. A modified preferred reporting items for systematic reviews and meta-analyses (PRISMA) diagram for the literature selection process is provided in Fig 1. A total of 7,090 citations were retrieved, with an additional 36 citations identified from checking the references of included studies. After removing duplicates, 4,803 citations remained. Of the 4,803 abstracts reviewed, 4,602 were excluded for clearly failing to meet at least one inclusion criterion or meeting at least one exclusion criterion, leaving 201 for full-text review. Of these, 41 were published only as abstracts and were excluded. Full texts of the remaining 160 citations were retrieved for detailed review. Of these, 40 papers reporting 44 WDMs satisfied the predefined inclusion criteria and were included in the review. The inter-reviewer agreement between HJ and XL, measured by Cohen's kappa was 0.58, which indicates moderate agreement. A list of excluded studies with reasons are reported in S1 Table. A list of identified pathways models is reported in S2 Table.

### Description of included WDMs

A brief summary of each included WDM is reported in Table 1. A summary of the key characteristics of the included WDMs are reported in Table 2. The most commonly modelled disease areas were heart disease (11/44, 25.0%), cancer (6/44, 13.6%), acquired immune deficiency syndrome (AIDS) (6/44, 13.6%) and metabolic disease (4/44, 9.1%). Of the 44 included WDMs, 33 (75.0%) met the criteria by demonstration and 11 (25.0%) met the criteria based on authors' reporting. Eleven WDMs (11/44, 25.0%) were developed using PopMod and have very similar characteristics [38]. PopMod is a standard modelling tool developed by the WHO-CHOICE programme to facilitate disease modelling and cost-effectiveness analysis in diverse settings. The tool uses a multi-state dynamic life table method to simulate the evolution in time of an arbitrary population subject to births, deaths and two distinct disease conditions. Within PopMod, the default time horizon is lifetime and the default effectiveness outcome is

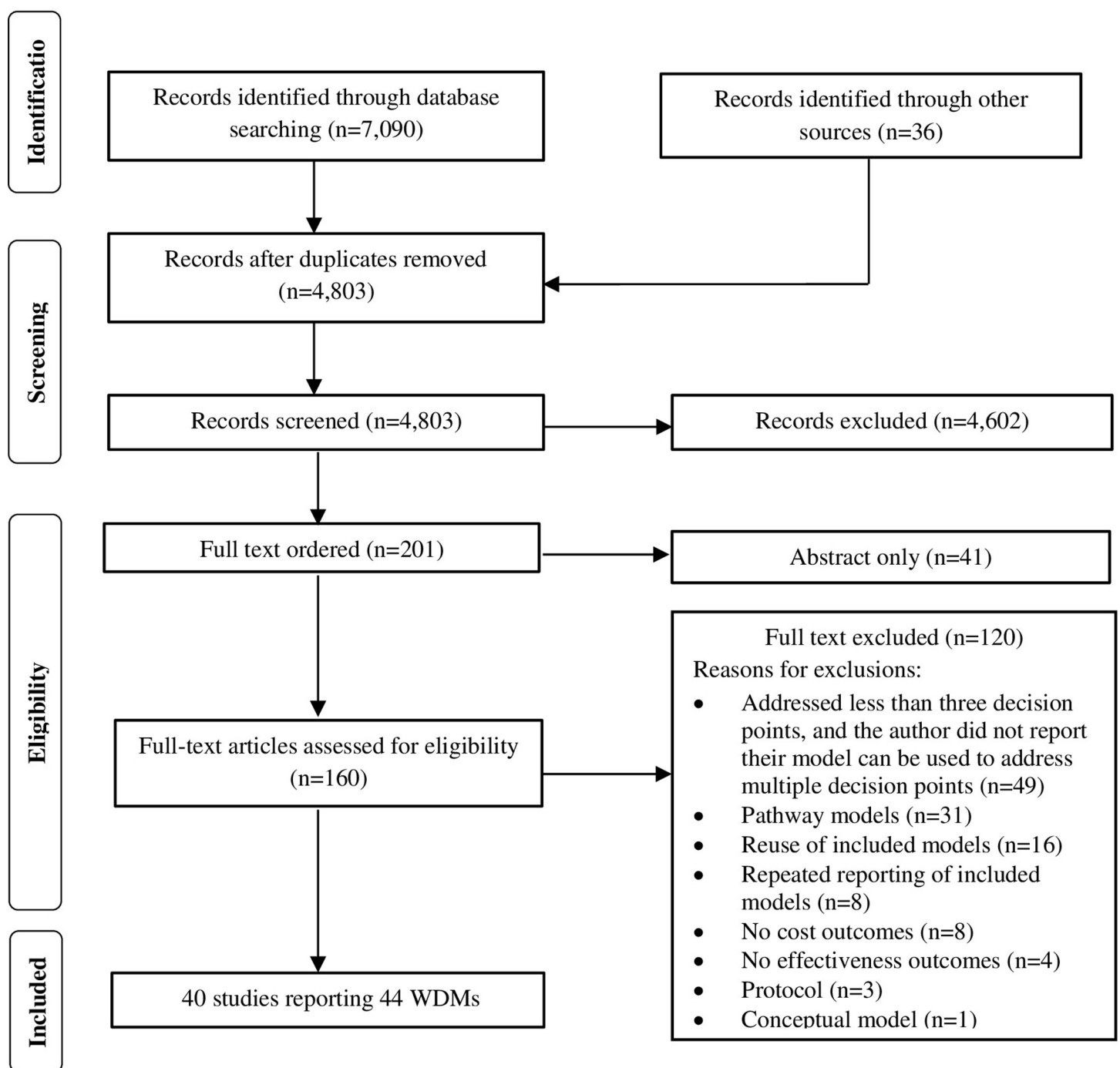

**Fig 1. Modified PRISMA flow diagram for the systematic review of economic models.**

disability-adjusted life years (DALYs). Of the eleven WDMs built using PopMod, eight addressed between 3 and 5 decision points (8/11, 72.7%) and three addressed between 6 and 10 decision points (3/11, 27.3%). Ten of the 11 PopMoD WDMs (90.9%) were published between 2010–2019. All eleven PopMoD WDMs were developed by non-commercial organisations. Five PopMoD WDMs (5/11, 45.5%) were developed for Sub-Saharan Africa and South

**Table 1.  Summary of included WDMs.**

| Disease areas | Author and reference | Year | Countries covered | Population | No. of decision points covered | Main effectiveness outcome | Perspective of cost | Modelling method |
|---|---|---|---|---|---|---|---|---|
| AIDS | Brandeau et al. [56] | 1991 | US | Population with differing risks for infection in the US | 1 | No. of infection and death | Societal | Dynamic compartmental model |
| AIDS | Juusola et al. [50] | 2016 | US | A population of adults including both HIV-infected and uninfected individuals. | 3 | QALY | Healthcare system | Not clearly reported |
| AIDS | Long et al. [57] | 2010 | US | High-risk (injection drug users, men who have sex with men) and low-risk individuals aged 15 to 64 in the U.S. | 3 | QALY | Societal | Dynamic compartmental model |
| AIDS | Stover et al. [45] | 2016 | Low- and middle-income countries (45 countries) | People who are at risk of, or infected with HIV | 19 | No. of HIV infections and death | Societal | Dynamic compartmental model |
| AIDS | Minnery et al. [51] | 2020 | Eswatini | People at risk of, or with a diagnosis of HIV in Eswatini | 4 | No. of HIV infections and death | Healthcare system | Dynamic compartmental model |
| AIDS | Seidu et al. [48] | 2021 | Ghana | People at risk of, or with a diagnosis of HIV in Ghana | 4 | No. of non-productive employees and HIV infections | Societal | Differential equation model |
| Alzheimer's disease | Kansal et al. [52] | 2018 | US | Patients with normal cognition or patients with different levels of dementia | 1 | QALYs | Societal | Individual patient-level model (not specified) |
| Cervical cancer | Ginsberg et al. [39] | 2012 | Sub-Saharan Africa and South East Asia | Women at risk of developing cervical cancer | 3 | DALY | Healthcare system | Multi-state dynamic life table |
| Cervical cancer | Ginsberg et al. [43] | 2009 | All 14 WHO regions | Women at risk of developing cervical cancer | 3 | DALY | Healthcare system | Multi-state dynamic life table |
| CHD | Davies et al. [28] | 2003 | UK | Adults aged 25 or more in the UK, with or without CHD | 3 | QALY | Healthcare | DES |
| CHD | Weinstein et al. [27] | 1987 | US | Adults aged 35 in the US, who have or have not developed CHD | 0 | Life years | Not reported | State transition model—cohort level |
| CHD in type 2 diabetes | Ye et al. [58] | 2015 | US | Patients with type 2 diabetes in the US, with and without CHD | 0 | QALYs | Not reported | State transition model—individual level |
| Colorectal cancer | Ginsberg et al. [39] | 2012 | Sub-Saharan Africa and South East Asia | Women at risk of developing colorectal cancer | 3 | DALY | Healthcare system | Multi-state dynamic life table |
| Colorectal cancer | Ginsberg et al. [44] | 2010 | All 14 WHO regions | Women at risk of developing colorectal cancer | 3 | DALY | Healthcare system | Multi-state dynamic life table |
| Colorectal cancer | Tappenden et al. [30] | 2013 | UK | General population with a normal epithelial state | 11 | QALY | Healthcare system | DES |
| COPD | Stanciole et al. [40] | 2012 | Sub-Saharan Africa and South East Asia | General population | 6 | DALY | Healthcare system | Multi-state dynamic life table |
| COPD | Hoogendoorn et al. [59] | 2011 | Netherlands | Dutch general population (each year a new birth cohort was added) and the COPD patient population in 2007 | 3 | QALY | Healthcare system | State transition model—cohort level |

*(Continued)*

**Table 1.** (Continued)

| Disease areas | Author and reference | Year | Countries covered | Population | No. of decision points covered | Main effectiveness outcome | Perspective of cost | Modelling method |
|---|---|---|---|---|---|---|---|---|
| COPD | Salomon et al. [42] | 2012 | Mexico | General population | 6 | DALY | Healthcare system | Multi-state dynamic life table |
| CVD | Basu et al. [60] | 2015 | India | Adults in India with risk factors for ischemic heart disease and cerebrovascular disease | 3 | DALY | Societal | State transition model—individual level |
| CVD | Ortegon et al. [41] | 2012 | Sub-Saharan Africa and South East Asia | People at risk of developing cardiovascular disease | 3 | DALY | Healthcare system | Multi-state dynamic life table |
| CVD | Salomon et al. [42] | 2012 | Mexico | General population | 3 | DALY | Healthcare system | Multi-state dynamic life table |
| CVD | Pandya et al. [61] | 2017 | US | Adult (ages 35–80 years) US general population with and without CVD | 0 | QALY | Not reported | State transition model—individual level |
| CVD and metabolic disease | Schlessinger and Eddy [26] | 2002 | US | People who are at risk of, or with diagnosed diabetes | 0 | Multiple, including incidence of diabetes and its complications and CHD events | Not reported | Ordinary differential equation |
| Depressive disorder | Lokkerbo et al. [54] | 2021 | The Netherlands | Dutch adults (aged 18–65 years) with subthreshold, mild, moderate, and severe major depression | 1 | QALY | Healthcare system | State transition model—cohort level |
| Heavy alcohol use | Salomon et al. [42] | 2012 | Mexico | General population | 5 | DALY | Healthcare system | Multi-state dynamic life table |
| Hypertension | Booth et al. [62] | 2007 | Finland | Individuals in Finland without diagnoses of diabetes, coronary heart disease, or cerebrovascular events, aged 40–74 | 3 | Life years | Healthcare system | State transition model—cohort level |
| Malaria | Edossa et al. [46] | 2023 | Ethiopia | People at risk of or with malaria | 4 | No. of infections averted | Not reported | Ordinary differential equation |
| Malaria | Goodman et al. [63] | 1999 | Sub-Saharan Africa | A hypothetical population with a life expectancy at birth of 50 years for very-low-income and middle-income countries, and a general pattern life table, with a life expectancy at birth of 65 years for higher-income countries | 3 | DALY | Healthcare system | Decision tree |
| Mental health disorders | Stelmach et al. [55] | 2022 | 36 low-income and middle-income countries | Adolescents (ages 10–19) from 36 countries at risk of, or with a diagnosis of anxiety, depression, bipolar disorder, and suicide | 7 | DALY | Societal | State transition model—cohort level |
| Multimorbidity: heart disease, Alzheimer's disease, and osteoporosis | Youn et al. [64] | 2019 | UK | General population aged 45 years and older | 3 | QALY | Healthcare system | DES |
| Myocardial infarction | Cretin [29] | 1977 | US | A cohort of 10-years-old males with confirmed or suspected myocardial infarction | 3 | Life years | Healthcare system | State transition model—cohort level |

*(Continued)*

**Table 1.** (Continued)

| Disease areas | Author and reference | Year | Countries covered | Population | No. of decision points covered | Main effectiveness outcome | Perspective of cost | Modelling method |
|---|---|---|---|---|---|---|---|---|
| Oral cancer | Cromwell *et al.* [65] | 2019 | Canada | Individuals who may or may not develop oral cancer | 5 | QALY | Healthcare system | DES |
| Osteoporosis | Hiligsmann *et al.* [66] | 2009 | Belgium | People at risk of developing osteoporosis | 1 | QALY | Healthcare system | State transition model—individual level |
| Pregnancy-related complications | Hu *et al.* [67] | 2007 | Mexico | Sexually active 15-year-old women at risk of becoming pregnant | 4 | DALY | Societal | State transition model—cohort level |
| Psychosis | Wijnen *et al.* [68] | 2020 | Netherlands | Individuals with ultra-high risk of developing psychosis or with first episode psychosis aged 25 years | 1 | QALY | Healthcare system | State transition model—cohort level |
| Retinopathy for patients with type 1 or 2 diabetes | Van Der Heijden *et al.* [69] | 2015 | Netherlands | Dutch population in 2003, consisting of persons with and without diabetes and its complications | 0 | QALY | Not reported | Dynamic compartmental model |
| Rheumatic Fever and Rheumatic Heart Disease | Watkins *et al.* [70] | 2016 | African Nations | General population | 3 | DALY | Healthcare system | State transition model—cohort level |
| Rheumatic heart disease | Coates et al. [47] | 2021 | African Union | People with a history of acute rheumatic fever (ARF) in the last 10 years (or under age 20), people with mild RHD, severe RHD (with HF), and RHD with valve replacement | 5 | Monetised health gains | Healthcare system | State transition model—cohort level |
| Schizophrenia | Jin *et al.* [31] | 2020 | UK | Individuals at risk of psychoses or with a diagnosis of psychosis or schizophrenia | 5 | QALY | Healthcare system | DES |
| Stroke | Mihalopoulos *et al.* [71] | 2005 | Australia | Australian population at risk of developing stroke | 4 | DALY | Societal | Not clearly reported |
| Tobacco | Salomon *et al.* [42] | 2012 | Mexico | General population | 5 | DALY | Healthcare system | Multi-state dynamic life table |
| Type 2 diabetes | Zhou *et al.* [53] | 2005 | US | People at risk of, or with a diagnosis of diabetes in the US | 0 | QALY | Healthcare system | State transition model—individual level |
| Type 2 diabetes | Sluijs *et al.* [72] | 2021 | The Netherlands | People at risk of, or with a diagnosis of type 2 diabetes in the Netherlands | 5 | No. of patents with type 2 diabetes | Societal | System Dynamics |
| Vision and hearing loss | Baltussen *et al.* [73] | 2012 | Sub-Saharan Africa and South East Asia | General population | 6 | DALY | Societal | Multi-state dynamic life table |

Abbreviations:

CHD = Coronary heart disease; COPD = Chronic obstructive pulmonary disease; CVD = Cardiovascular disease; DALY = disability-adjusted life years; DES = discrete event simulation; QALY = quality-adjusted life years; WDM = whole disease model.

East Asia [39–41], four (36.4%) were developed for Mexico [42] and two (18.2%) were developed for all 14 WHO regions [43, 44]. All eleven PopMoD WDMs used the piecewise cost per DALY threshold decision rule. None of the PopMod WDMs applied a disease-level constrained maximisation decision rule.

The characteristics of the remaining 33 WDMs varied greatly: for the sake of clarity, models which did not use the PopMod method are hereafter referred to as "other WDMs" throughout

**Table 2. Characteristics of included WDMs.**

| | PopMod WDMs (n = 11) n (%) | Other WDMs (n = 33) n (%) | Total (n = 44) n (%) |
|---|---|---|---|
| **Met the WDM criteria by demonstration or authors' reporting** | | | |
| Demonstration | 11 (100.0) | 22 (66.7) | 33 (75.0) |
| Authors' reporting | 0 (0.0) | 11 (33.3) | 11 (25.0) |
| **Disease area [a]** | | | |
| Addiction (e.g., tobacco and alcohol) | 2 (18.2) | 0 (0.0) | 2 (4.5) |
| AIDS | 0 (0.0) | 6 (18.2) | 6 (13.6) |
| Alzheimer's disease | 0 (0.0) | 2 (6.1) | 2 (4.5) |
| Bone disease | 0 (0.0) | 2 (6.1) | 2 (4.5) |
| Cancer | 4 (36.4) | 2 (6.1) | 6 (13.6) |
| COPD | 2 (18.2) | 1 (3.0) | 3 (6.8) |
| Depressive disorder | 0 (0.0) | 1 (3.0) | 1 (2.3) |
| Eye disease | 1 (9.1) | 1 (3.0) | 2 (4.5) |
| Heart disease | 2 (18.2) | 9 (27.3) | 11 (25.0) |
| Hearing loss | 1 (9.1) | 0 (0.0) | 1 (2.3) |
| Malaria | 0 (0.0) | 2 (6.1) | 2 (2.3) |
| Metabolic disease | 0 (0.0) | 4 (12.1) | 4 (9.1) |
| Multiple mental health disorders | 0 (0.0) | 1 (3.0) | 1 (2.3) |
| Pregnancy-related complications | 0 (0.0) | 1 (3.0) | 1 (2.3) |
| Psychosis/schizophrenia | 0 (0.0) | 2 (6.1) | 2 (4.5) |
| Rheumatic heart disease | 0 (0.0) | 1 (3.0) | 1 (2.3) |
| Rheumatic fever and rheumatic heart disease | 0 (0.0) | 1 (3.0) | 1 (2.3) |
| **Number of decision points addressed in the paper** | | | |
| 0–2 [b] | 0 (0.0) | 11 (33.3) | 11 (25.0) |
| 3–5 | 8 (72.7) | 19 (56.3) | 27 (61.4) |
| 6–10 | 3 (27.3) | 1 (3.0) | 4 (9.1) |
| Over 10 | 0 (0.0) | 2 (6.1) | 2 (4.5) |
| **Year of publication** | | | |
| Before 2000 | 0 (0.0) | 4 (12.1) | 4 (9.1) |
| 2000–2009 | 1 (9.1) | 7 (21.2) | 8 (18.2) |
| 2010 onwards | 10 (90.9) | 22 (65.6) | 32 (72.7) |
| **Countries covered by the model [c]** | | | |
| Sub-Saharan Africa and South East Asia | 5 (45.5) | 1 (3.0) | 6 (13.6) |
| Mexico | 4 (36.4) | 1 (3.0) | 5 (11.4) |
| All 14 WHO regions | 2 (18.2) | 0 (0.0) | 2 (4.5) |
| USA | 0 (0.0) | 10 (30.3) | 10 (22.7) |
| Netherlands | 0 (0.0) | 5 (15.2) | 5 (11.4) |
| UK | 0 (0.0) | 4 (12.1) | 4 (9.1) |
| Other | 0 (0.0) | 12 (36.4) | 12 (27.3) |
| **Type of economic evaluation** | | | |
| Cost-utility analysis [d] | 11 (100.0) | 21 (63.6) | 32 (72.7) |
| Cost-effectiveness analysis | 0 (0.0) | 11 (31.3) | 11 (25.0) |
| Cost-benefit analysis | 0 (0.0) | 1 (3.0) | 1 (2.3) |
| **Main effectiveness outcome** | | | |
| QALY | 0 (0.0) | 16 (48.5) | 16 (36.4) |
| DALY | 11 (100.0) | 6 (18.2) | 17 (38.6) |
| Disease incidence | 0 (0.0) | 5 (15.2) | 5 (11.4) |
| Life years | 0 (0.0) | 3 (9.1) | 3 (6.8) |

(*Continued*)

**Table 2.** (Continued)

| | PopMod WDMs (n = 11) n (%) | Other WDMs (n = 33) n (%) | Total (n = 44) n (%) |
|---|---|---|---|
| Monetised health gains | 0 (0.0) | 1 (3.0) | 1 (2.3) |
| Number of infections averted | 0 (0.0) | 1 (3.0) | 1 (2.3) |
| No. of non-productive employees | 0 (0.0) | 1 (3.0) | 1 (2.3) |
| **Perspective of cost** | | | |
| Healthcare system | 10 (90.9) | 17 (51.5) | 27 (61.4) |
| Society | 1 (9.1) | 10 (30.3) | 11 (25.0) |
| Not reported | 0 (0.0) | 6 (15.6) | 6 (13.6) |
| **Time horizon** | | | |
| <10 year | 0 (0.0) | 2 (6.1) | 2 (4.5) |
| 10–30 year | 0 (0.0) | 10 (30.3) | 10 (22.7) |
| 31–80 year | 0 (0.0) | 5 (15.2) | 5 (11.4) |
| Lifetime | 11 (100.0) | 14 (42.4) | 25 (56.8) |
| Not reported | 0 (0.0) | 2 (3.1) | 2 (4.5) |
| **Modelling techniques adopted** | | | |
| Multi-state dynamic life table | 11 (100.0) | 0 (0.0) | 11 (25.0) |
| State transition model—cohort level | 0 (0.0) | 10 (30.3) | 10 (22.7) |
| State transition model—individual level | 0 (0.0) | 5 (15.2) | 5 (11.4) |
| Discrete event simulation | 0 (0.0) | 5 (15.2) | 5 (11.4) |
| Dynamic compartmental model | 0 (0.0) | 5 (15.2) | 5 (11.4) |
| Decision tree | 0 (0.0) | 1 (3.0) | 1 (2.3) |
| Ordinary differential equation | 0 (0.0) | 2 (6.1) | 2 (4.5) |
| Differential equation model | 0 (0.0) | 1 (3.0) | 1 (2.3) |
| System dynamics | 0 (0.0) | 1 (3.0) | 1 (2.3) |
| Not clearly reported | 0 (0.0) | 3 (9.1) | 3 (6.8) |
| **Modelling software** | | | |
| WHO PopMod | 11 (100.0) | 0 (0.0) | 11 (25.0) |
| SIMUL8 | 0 (0.0) | 3 (9.1) | 3 (6.8) |
| Excel | 0 (0.0) | 3 (9.1) | 3 (6.8) |
| Python | 0 (0.0) | 3 (9.1) | 3 (6.8) |
| TreeAge | 0 (0.0) | 2 (6.1) | 2 (4.5) |
| R | 0 (0.0) | 2 (6.1) | 2 (4.5) |
| Mathematica | 0 (0.0) | 2 (6.1) | 2 (4.5) |
| Other | 0 (0.0) | 10 (28.1) | 10 (22.7) |
| Not reported | 0 (0.0) | 8 (24.2) | 8 (18.2) |
| **Decision rule supported by the model** [e] | | | |
| Piecewise cost per additional unit of effectiveness outcome threshold rule | 11 (100.0) | 33 (100.0) | 44 (100.0) |
| Disease-level constrained maximisation of effectiveness outcome decision rule | 0 (0.0) | 4 (12.1) | 4 (9.1) |
| **Access to the model** | | | |
| Open access | 0 (0.0) | 5 (15.2) | 5 (11.4) |
| Not reported | 11 (100.0) | 28 (84.8) | 39 (88.6) |
| **Affiliation of corresponding author** | | | |
| Commercial | 0 (0.0) | 3 (9.1) | 3 (6.8) |

(*Continued*)

**Table 2.** (Continued)

| | PopMod WDMs (n = 11) n (%) | Other WDMs (n = 33) n (%) | Total (n = 44) n (%) |
|---|---|---|---|
| Non-commercial | 11 (100.0) | 30 (90.6) | 41 (93.2) |

Abbreviations:

AIDS = acquired immune deficiency syndrome; COPD = chronic obstructive pulmonary disease; DALY = disability-adjusted life year; QALY = Quality-adjusted life year; WDM = whole disease model.

Notes:

a. Four studies covered more than one disease area [26, 55, 64, 73].

b. A model addressed less than two decision points in the paper can still be considered to a WDM if the authors clearly reported their model can be used to address more than 2 decision points.

c. Twelve studies covered more than one country.

d. Within a cost-utility analysis, effectiveness is measured either as QALYs or DALYs.

e. Four studies supported more than one decision rule [30, 31, 50, 51].

this paper. Eleven addressed less than 3 decision points but reported that they could address 3 or more (11/33, 33.3%), 19 addressed 3–5 decision points (57.6%) and 2 WDMs addressed over 10 decision points (6.1%) [30, 45]. These models were published between 1977 [29] to 2023 [46]. 66.7% of these WDMs (22/33) were published from 2010 onwards. The time horizon adopted ranged from 10-years to a lifetime. Twenty-one studies were cost-utility analyses (21/33, 63.6%), 11 studies were cost-effectiveness analyses (11/33, 33.3%) and one study was cost-benefit analysis (1/33, 3.0%) [47]. Sixteen studies used QALYs as the main effectiveness outcome (16/33, 48.5%), the rest used DALYs (6/33, 18.2%), disease incidence (5/33, 15.2%), life years (3/33, 9.1%), monetised health gains (1/33, 3.0%) [47], number of infection cases prevented (1/33, 3.0%) [46], and number of non-productive employees (1/33, 3.0%) [48]. The most commonly adopted modelling methods were cohort-level state transition model (10/33, 30.3%), individual-level state transition model (5/33, 15.2%), discrete event simulation (DES) (5/33, 15.2%), and dynamic compartmental model (5/33, 15.2%). The modelling software packages used were SIMUL8 (3/33, 9.1%), Excel (3/33, 9.1%), Python (3/33, 9.1%), TreeAge (2/33, 6.1%), R (2/33, 6.1%), Mathematica (2/33, 6.1%) and other (9/33, 27.3%). Eight WDMs did not report the modelling software used (8/33, 24.2%). Most of the WDMs were developed for high income countries, such as the USA (10/33, 30.3%), Netherlands (5/33, 15.2%), and the UK (4/33, 12.1%). All the WDMs compared competing options using a piecewise cost-effectiveness threshold-based decision rule; four of which (12.1%) also used disease-level constrained maximisation decision rule [31, 49–51]. Three of the WDMs were developed by commercial organisations [26, 45, 52] (3/33, 9.1%); the remainder were developed by non-commercial organisations (30/33, 90.9%).

Of all 44 identified WDMs, five reported that their models can be downloaded for free (11.4%) [47, 51, 53–55]. The other 39 WDMs (88.6%) did not report whether their model can be accessed by other researchers or not.

The complete evidence table for all included WDMs is reported in S3 Table.

## Quality assessment

The results of the quality assessment for WDMs are summarised below; further detail is provided in the S4 Table. The reporting of most included WDMs was poor. Nineteen WDMs (19/44, 43.2%), including all eleven PopMod WDMs did not report details of their model structure or present their model structure diagrammatically. Fifteen WDMs (15/44, 34.1%) did not report the evidence sources used to inform resource use or unit cost inputs, and ten WDMs

(10/44, 22.7%) did not report the evidence sources used to inform model parameters relating to clinical effectiveness of the technologies used within the treatment pathways.

According to the quality assessment results of the NICE checklist, of all 44 WDMs, 35 were deemed to have very serious limitations (79.5%), including all PopMoD models; five were deemed to have potentially serious limitations (11.4%) [47, 54, 55, 59, 71]; and four were deemed to have minor limitations (9.1%) [30, 31, 61, 65]. The performance of included Pop-Mod WDMs (n = 11), other WDMs (n = 33), and all WDMs (n = 44) on all items of the NICE checklist is shown in Fig 2A–2C, respectively.

Of the PopMoD models, all of which were deemed to have very serious limitations, key problems identified included (Fig 2A):

1. Inadequate model structure (11/11, 100%): PopMod is aimed at modelling initial disease treatment only and does not consider relapse or progress status; therefore, it is unlikely to capture the complexity of natural disease history or the entire care pathway.

2. Failure to include all important and relevant outcomes and costs (11/11, 100%): the standard PopMod model only has four health states, including two groups with specific disease conditions, a group with the combined condition and a group with neither condition; as a result, PopMod has very limited capacity in modelling the health impacts of any adverse events (AEs) of interventions. None of the eleven PopMod WDMs in this review modelled any treatment-related AEs or justified their exclusion.

3. Insufficient sensitivity analyses, for example, deterministic analysis for less than three parameters, or lack of probabilistic sensitivity analysis (PSA): PSA is useful for assessing the impact of joint uncertainty of multiple parameters simultaneously. However, the current version of PopMod does not support PSA and of the eleven PopMod WDMs included, only four of them conducted PSA [39, 41, 74, 75], and this required the use of an additional software package (MCLeague).

Of the 33 other WDMs, 24 were deemed to have very serious limitations (72.7%), five were deemed to have potentially serious limitations (15.2%), and four were deemed to have minor limitations (12.1%). Four key problems were identified (Fig 2B). First, twenty-nine WDMs (87.9%) failed to include all important and relevant outcomes and costs, for example, dis-utilities and costs resulting from AEs of interventions. Second, twenty-two WDMs (66.7%) failed to explore important uncertainties in their analysis using PSA or only conducted one-way sensitivity analyses for less than three parameters. Third, twenty WDMs (60.6%) did not report the source of resource use data or did not obtain resource use data from the best available source. For example, resource use data were estimated based on expert opinion or obtained from countries other than the one(s) of interest. Fourth, eighteen WDMs (54.5%) did not report their source of unit cost data or did not obtain unit cost data from the best available source. For example, unit cost data were estimated based on expert opinion or obtained from countries other than the one(s) of interest.

## Discussion

### Main findings and interpretation

Our review identified 44 WDMs. The first WDM was published as early as 1977 [29]. Over 70% of WDMs were published after 2010. The main disease areas covered by existing WDMs are heart disease, cancer, AIDS and metabolic disease, all of which are associated with significant disease burden. Only three WDMs (8%) were developed by commercial organisations [26, 45, 52]. This might be because most commercial companies are more likely to be

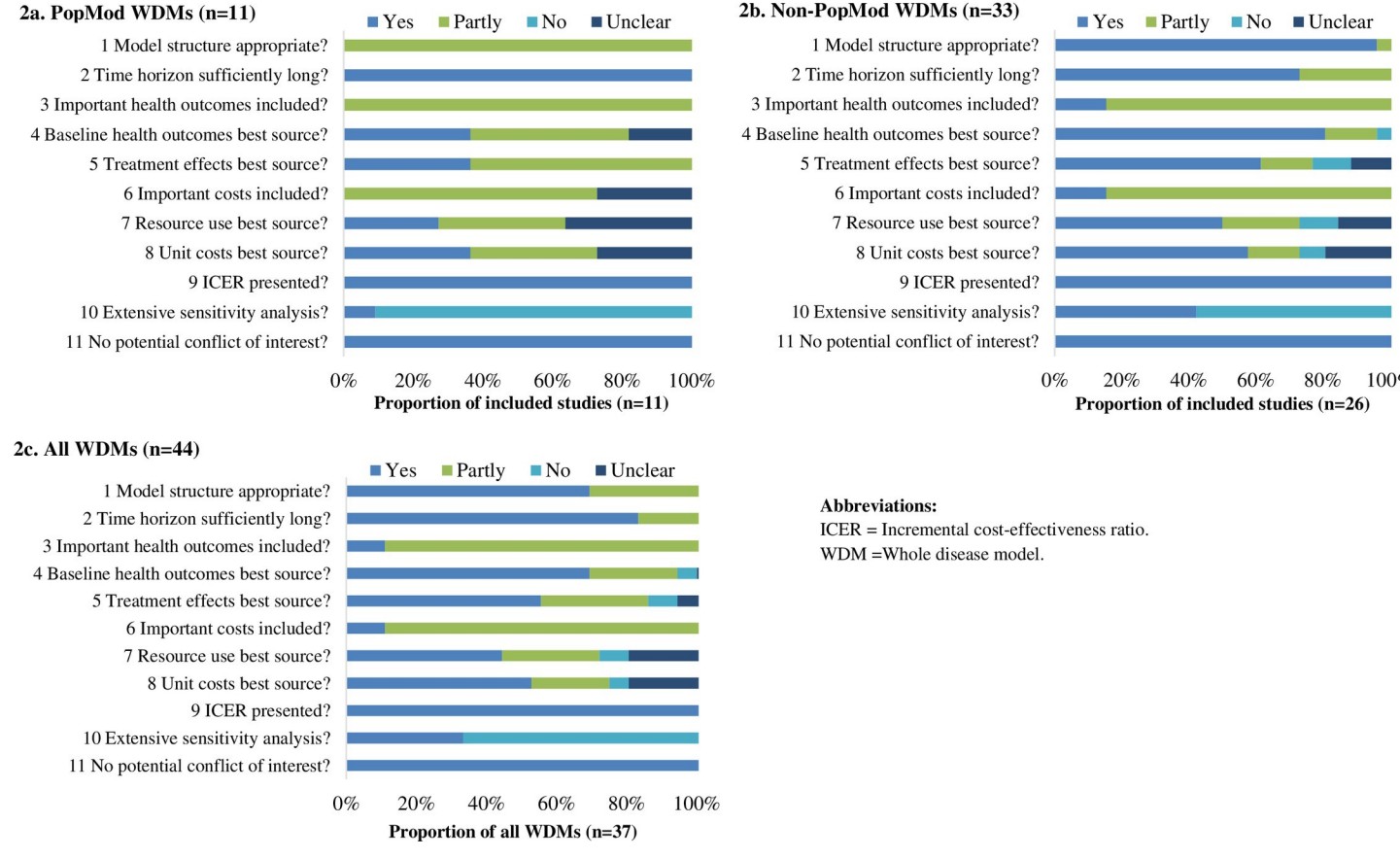

**Fig 2. Performance of included studies assessed by Section 2 of the NICE checklist.**

motivated to develop piecewise models which evaluate a specific (type of) intervention, rather than developing WDMs which covers all interventions across the entire care pathway.

The majority of WDMs were of poor quality which means they may require significant modification before they can be re-used. These limitations included failure to consider the dis-utilities and cost caused by AEs of interventions, and lack of PSA. It was estimated that in OECD countries, 15% of total hospital activity and expenditure was a direct result of AEs [76]. Worldwide, the total disability-adjusted life years (DALYs) due to AEs of medical treatment was estimated to be 62.8 per 100,000 population in 2017 [77]. It has been reported that, for patients with schizophrenia [25], the cost-effectiveness of an intervention tends to be driven by its AE profile, rather than its clinical effectiveness. Given the potentially substantial impact of adverse events on cost-effectiveness results, many checklists and guidance for health economic analysis, such as the NICE checklist [37], the Philips's checklist [36], Cooper's hierarchy [78] and the ISPOR good practice guidance for budget impact analysis [79], all recommend that the dis-utilities and/or cost caused by AEs of interventions need to be considered in health eco-nomic analyses. PSA is mandated by many health technology assessment agencies, including the UK's National Institute for Health and Care Excellence (NICE) [1] and the Canadian Agency for Drugs and Technologies in Health (CADTH) [80]. PSA is also recommended in the guidelines produced by the International Society for Pharmacoeconomics and Outcomes Research (ISPOR) [81]. This is because for non-linear decision models (i.e. the relationship between the input data and the outcomes is not linear), PSA is required to provide a reliable

estimate of mean costs and outcomes [1]. Given the complexity of entire disease pathways, WDMs are likely to be non-linear models. Therefore, being able to run PSA is an important function of a WDM.

Other limitations which may reduce the possibility or value of the WDMs for future use included (i) no reporting of whether their models can be accessed by other researchers or not; (ii) choice of modelling method and (iii) choice of modelling software. Of all WDMs included, only 23% used an individual-level modelling method. In order to evaluate multiple decision questions across the entire care pathway, a WDM often needs to incorporate a large number of events of interest and the relevant risk factors for all events of interest. If a cohort-level model was used to represent a large number of events and risk factors, thousands or even millions of health states will need to be built, which may place a huge burden on the model development and validation [82, 83]. Compared with cohort-level models, individual-level models might be more appropriate for developing a WDM, because they allow individual patient characteristics (e.g., risk factors for events of interest) and histories (e.g., events of interest) to be recorded [83–85]. In addition, individual-level models allow patients' risk of events to change over time, depending on patient characteristics and previous events. Therefore, the methodological framework for developing WDMs [9] suggest that 'given the level of depth and flexibility required to transfer the decision node across the model pathway, it is highly likely that individual-level simulation will be required." One drawback of individual-level models is they require more computation time to run PSA compared with cohort-level models [86–88]. However, as computing power increases over time, the difference in computational burden between cohort-level and individual-level models reduces significantly [82, 83].

In terms of modelling software, our review found that a quarter of WDMs were developed using PopMoD. The main benefits of using PopMod are that such models are easy to build, free to use, allow modelling of one comorbidity, and support separate modelling of age and time [38]. In addition, the use of a standard modelling template facilitates comparison of results across different disease areas. However, PopMoD may not be the ideal tool in the context of building a WDM. Due to its simple structure with only four health states, PopMod does not allow modelling of disease relapse or progression or modelling of AEs, both of which are important considerations for a WDM which covers the entire natural disease history and a range of interventions. Other WDMs included in the review were developed using various modelling software, most of which are proprietary specialist software, such as Simul8, TreeAge and Matlab. Compared with open-source coding languages such as R and Python, specialist software might be easier to learn and use as they are equipped with pre-defined modules/functions. However, these software packages are often associated with a license fee.

### Use of WDMs in clinical guidelines

By covering the entire care pathway and most key interventions within a single modelling framework, whole disease modelling can be used as an ideal foundation for economic evaluations in clinical guidelines. So far, whole disease modelling has been piloted on three published NICE clinical guidelines, covering three different disease areas: colorectal cancer [89], prostate cancer [90] and atrial fibrillation [91]. The reported advantages of using whole disease modelling for the selected NICE guidelines are: (a) it produces a considerably larger amount of economic information compared to traditional 'piecewise' models; (b) it improves consistency of cost-effectiveness results by using a single analytic framework and a common set of assumptions and data sources; (c) it can be used to compare the cost-effectiveness of interventions at different parts of the pathway (e.g. prevention and treatment), and explore system

interdependencies between different interventions; and (d) once developed, WDMs can be reused to consider other related questions or to incorporate new evidence [30, 32].

The reported disadvantages of using whole disease modelling are mainly related to the technical and practical barriers. Large and complex models like WDMs are likely to be more prone to verification (programming) errors, more difficult to validate, and more difficult to explain to decision-makers than simple models. As a result, the development of each WDM has been estimated to take at least 12 months' time of a full-time modeller. However, as Professor Alan Williams [92] pointed out, if creating such large-scale models is horrendously complicated, it is because reality is horrendously complicated; the more complex the reality is, the more dangerous it is to rely on intuitive short-cuts rather than careful analyses. Therefore, 'more complex areas require models that respect complexity' [84]. Given that WDMs, once developed, can be adapted and reused to provide ongoing support for decision-making across the entire care pathway, in the long-term, the benefits of using whole disease modelling in the context of clinical guidelines may outweigh the initial investment of time and money.

### Recommendations for future research

Based on the findings of this review, there are a number of recommendations for future research:

1. The reporting of future WDMs should follow the CHEERS statement [35]. In addition to the items listed in the CHEERS statement, we recommend that the following information should also be reported to help other researchers to decide whether the WDM can be reused/adapted to their settings: (1) software used to develop the model; and (2) a statement about access to the model, e.g. whether their model can be accessed by other researchers, and under what conditions. It is also recommended that a user manual is developed which includes a detailed description of each component of the model structure, all input data and instructions on how to use and/or adapt the model, as recommended by the STRESS guidelines for reporting models [93].

2. It is recommended that for future WDMs, the appropriateness of alternative modelling methods should be assessed and the chosen method justified, this can be achieved by the application of model selection tools, such as the revised Brennan *et al* taxonomy [82].

3. The health and cost impacts of important AEs of interventions should be considered for inclusion in the model and only excluded where these impacts are negligible and a clear justification is provided.

4. Extensive sensitivity analyses need to be conducted to explore all key uncertainties in the model. As a minimum, one-way sensitivity analysis for all key parameters and PSA should be conducted.

In addition to the recommendation listed above, future research exploring methods for reducing the development time of WDMs is required to encourage greater use of the whole disease modelling approach. There are two solutions which could potentially help to reduce the development time of a WDM. The first solution is to adopt a team approach. The development of a WDM can be divided into several interrelated but different task modules, each of which requires a different set of skills (e.g., development of a conceptual model, identification and preparation of relevant input data, implementing the conceptual model within a computer software, and communicating the model results to stakeholders). This means the development work for a WDM can be potentially assigned to more than one researcher working in parallel, which would accelerate the development time of a WDM. However, the adoption of a team

approach will lead to increased labour costs. The second solution is to develop templates for whole disease modelling. There are some similarities in terms of building health economic models for diseases of the same or similar types. For example, the modelling of cancer usually involves tumour progression from early stages to late stages, while the modelling of mental health problems usually involves repeated transitions between remission and relapse. Even for patients with different types of diseases, their QALYs accumulated at a specific stage of the care pathway usually depend on similar factors, such as: patient's starting age, end age, current disease status, comorbidities and/or adverse events of interventions. The code for implementing these common functions can be made into a modelling template (or different templates for different types of disease) and published online. These templates may help to reduce the development time of future WDMs.

## Strengths & limitations

**Strengths.**   To our knowledge, this is the first systematic review which outlines the availability and quality of WDMs for any disease areas. The information reported by this systematic review can be used to help researchers, commissioners or other stakeholders to rapidly locate relevant WDMs in the disease areas that they are interested in and critically appraise existing WDMs. Recommendations for future research can be used to fill evidence gaps and improve the quality and reporting of future WDMs.

**Limitations.**   This review is subject to at least five main limitations. Firstly, our search might not have identified all models which meet the criteria for a WDM. When designing the search strategy, we tried to include any search terms which might be relevant to a WDM, including terms such as "(full/comprehensive/entire/whole) adj3 (pathway*/system*/guideline*/disease*)", "upstream adj2 downstream", and "prevent* adj7 treat*". However, unless we searched for all models for any diseases, there is no guarantee that all models meeting the criteria of a WDM have been identified. In addition, non-English literature and grey literature were not searched due to constraints in time, resources, and expertise within the reviewing team for non-English languages; and the inter-reviewer agreement is moderate (Cohen's kappa = 0.58). The two reviewers (HJ and XL) tried to be more inclusive rather than exclusive when discussing any disagreed studies, however, there is still a possibility that some relevant WDMs have been missed by our review. Since the aim of this review was to provide an overview of existing WDMs, rather than to synthesise the results of the identified studies, we reckon the negative consequences of missing relevant studies are relatively small. Secondly, it should be noted that the definition of a WDM used for this review is more inclusive than the original definition provided in Tappenden *et al.* [9]. For example, Tappenden *et al.* defines a WDM as a model which includes the entire preclinical and post-diagnostic pathways for a given disease, allows the use of disease-level constrained optimisation, and allows decision node to be transferred across the modelled pathway. Since models meeting such criteria are rare, we decided to relax the definition of a WDM for this review to include models which can evaluate multiple decision points covering both the prevention and treatment of the disease simultaneously. This means not all WDMs included in this review cover the entire preclinical and post-diagnostic pathways for a given disease. As a result, not all included WDMs allow the use of disease-level constrained optimisation–of the 43 WDMs included, only four [31, 49–51] demonstrated that they allow the use of disease-level constrained optimisation. In addition, those included WDMs which were developed using a cohort-level modelling method are unlikely to allow a decision node to be transferred across the modelled pathway. Thirdly, whether a paper meets our inclusion criteria or not was determined based on the content of each individual paper, rather than the content of a series of related papers. Therefore, we

might have missed those models which were used to address multiple decision points in a series of papers, but each paper only used that model to address one or two decision points. Fourthly, for those models which claimed they can be used to address three or more decision points but did not demonstrate this in the paper, whether they met the inclusion criteria of a WDM/pathway model or not was based on the authors' reporting rather than our own assessment. Finally, we used the NICE checklist for economic evaluations [37] for assessing the quality of included WDMs. However, it should be noted that not all important aspects of quality assessment, such as model performance (e.g. comparing model results to real-world results) or model validation activities, were covered by the NICE checklist. In addition, the importance of quality criteria that a study fails (i.e. how likely this will change the conclusions about cost-effectiveness) was based on the reviewer's judgement. Therefore, our results of quality assessment need to be interpreted with caution.

## Conclusion

Despite their significant resource requirements associated with model development, there has been a significant increase in the number of WDMs since 2010. The main disease areas covered by existing WDMs are heart disease, cancer, metabolic disease and AIDS. A quarter of included WDMs were multi-state dynamic life table models developed using PopMoD; the remaining WDMs were developed using various modelling methods and software. The majority of WDMs were of poor quality which means they may require significant modification before they can be re-used, such as modelling AEs of interventions and incorporation of PSA. It is recommended that sufficient details of the WDMs need to be reported to allow future reuse/adaptation.

## Supporting information

**S1 Text. Electronic search strategies.**
(DOCX)

**S1 Table. List of excluded studies with reasons.**
(XLSX)

**S2 Table. Characteristics of included pathway models.**
(XLSX)

**S3 Table. Characteristics of included whole disease models.**
(XLSX)

**S4 Table. Quality assessment result.**
(XLSX)

**S1 Checklist. PRISMA 2009 checklist.**
(DOC)

## Author Contributions

**Conceptualization:** Huajie Jin, Paul Tappenden, Sarah Byford.

**Data curation:** Huajie Jin, Xiaoxiao Ling.

**Formal analysis:** Huajie Jin, Xiaoxiao Ling.

**Methodology:** Huajie Jin, Paul Tappenden, Stewart Robinson, Sarah Byford.

**Project administration:** Huajie Jin.

**Software:** Huajie Jin.

**Supervision:** Huajie Jin, Paul Tappenden, Stewart Robinson, Sarah Byford.

**Validation:** Huajie Jin.

**Writing – original draft:** Huajie Jin, Paul Tappenden, Sarah Byford.

**Writing – review & editing:** Huajie Jin, Paul Tappenden, Xiaoxiao Ling, Stewart Robinson, Sarah Byford.

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
