## [Decision Letter · Decision Letter 0]

9 Nov 2022

PONE-D-22-22167A systematic review of whole disease models for informing healthcare resource allocation decisionsPLOS ONE

Dear Dr. Jin,

Thank you for submitting your manuscript to PLOS ONE. After careful consideration, we feel that it has merit but does not fully meet PLOS ONE’s publication criteria as it currently stands. Therefore, we invite you to submit a revised version of the manuscript that addresses the points raised during the review process.

Reviewers' comments:

Reviewer's Responses to Questions

**Comments to the Author**

1. Is the manuscript technically sound, and do the data support the conclusions?

Reviewer #1: Yes

Reviewer #2: Yes

2. Has the statistical analysis been performed appropriately and rigorously? 

Reviewer #1: Yes

Reviewer #2: Yes

3. Have the authors made all data underlying the findings in their manuscript fully available?

Reviewer #1: Yes

Reviewer #2: Yes

4. Is the manuscript presented in an intelligible fashion and written in standard English?

Reviewer #1: Yes

Reviewer #2: Yes

5. Review Comments to the Author

Reviewer #1: This systematic review identified and summarized the characteristics of existing whole disease models (WDMs), and critically assessed their quality using the NICE appraisal checklist for economic evaluations. The information reported by this article can be used to help researchers, commissioners, or other stakeholders to rapidly locate relevant WDMs in the disease areas that they are interested in and critically appraise existing WDMs. Recommendations for future research can be used to fill evidence gaps and improve the quality and reporting of future WDMs. This article is well written, but I have the following concerns about the manuscript.

1. The definition of “disease-level constrained maximisation of health decision rule” in this article was ambiguous, please use more straightforward words to clarify.

2. Lines 129-130, the subtitle of “Assessment of abstracts for inclusion” was inconsistent with the text of “Screening of abstracts and papers against the inclusion criteria...”. Please unify the descriptions.

3. Lines 203, “The characteristics of the remaining 26 WDMs...” should be “The characteristics of the remaining 32 WDMs...”.

4. Table 2, the total number of items in the “Modelling techniques adopted” and “Modelling software” sections was not equal to the total number of included WDMs (n=43). This issue can also be found in the corresponding text (lines 212-216). Please check and correct them.

5. Line 378, “the inter-reviewer agreement is moderate (Cohen's kappa = 0.56)” was inconsistent with the statement of “Cohen's kappa was 0.58” in the result section (line 180). Please double-check the figures.

6. Lines 390-391, the data in the sentence “of the 37 WDMs included, only three demonstrated that they allow the use of disease-level constrained optimisation” needs to be updated to reflect “the update search”.

7. It is suggested that the review further reflect on the impact of the development and application of WDMs on disease management and clinical pathway optimization, and then provide recommendations on the content and direction for future WDM construction.

Reviewer #2: A systematic review of whole disease models for informing healthcare resource

allocation decisions

There has been an increase in the number of WDMs since 2010. However, their quality is generally

low which means they may require significant modification before they could be re-used, such as modelling AEs of

interventions and incorporation of PSA. Sufficient details of the WDMs need to be reported to allow future

reuse/adaptation 

6. PLOS authors have the option to publish the peer review history of their article (what does this mean?). If published, this will include your full peer review and any attached files.

**Do you want your identity to be public for this peer review?** For information about this choice, including consent withdrawal, please see our Privacy Policy.

Reviewer #1: No

Reviewer #2: **Yes: **Mohnad Abdalla

We look forward to receiving your revised manuscript.

Kind regards,

Nyanyiwe Masingi Mbeye, Ph.D

Academic Editor

PLOS ONE
---

## [Author Response · Author response to Decision Letter 0]

11 Nov 2022

Please see uploaded response letter.

---

## [Decision Letter · Decision Letter 1]

25 Apr 2023

PONE-D-22-22167R1A systematic review of whole disease models for informing healthcare resource allocation decisionsPLOS ONE

Dear Dr  Jin,

Thank you for submitting your manuscript to PLOS ONE. After careful consideration, we feel that it has merit but does not fully meet PLOS ONE’s publication criteria as it currently stands. Therefore, we invite you to submit a revised version of the manuscript that addresses the points raised during the review process.

We look forward to receiving your revised manuscript.

Kind regards,

Karina Cardoso Meira, Ph.D

Academic Editor

PLOS ONE

Journal Requirements:

Additional Editor Comments (if provided):

Thank you for submitting your manuscript to PLOS ONE. After careful consideration, we feel that it has merit but does not fully meet PLOS ONE’s publication criteria as it currently stands. Therefore, we invite you to submit a revised version of the manuscript that addresses the points raised during the review process.

There are few weaknesses in the methods that must be addressed in full before the article can be reconsidered for publication.

Reviewers' comments:

Reviewer's Responses to Questions

**Comments to the Author**

1. If the authors have adequately addressed your comments raised in a previous round of review and you feel that this manuscript is now acceptable for publication, you may indicate that here to bypass the “Comments to the Author” section, enter your conflict of interest statement in the “Confidential to Editor” section, and submit your "Accept" recommendation.

Reviewer #2: All comments have been addressed

Reviewer #3: All comments have been addressed

2. Is the manuscript technically sound, and do the data support the conclusions?

Reviewer #2: Yes

Reviewer #3: Yes

3. Has the statistical analysis been performed appropriately and rigorously? 

Reviewer #2: Yes

Reviewer #3: N/A

4. Have the authors made all data underlying the findings in their manuscript fully available?

Reviewer #2: Yes

Reviewer #3: Yes

5. Is the manuscript presented in an intelligible fashion and written in standard English?

Reviewer #2: Yes

Reviewer #3: Yes

6. Review Comments to the Author

Reviewer #2: A systematic review of whole disease models for informing healthcare resource allocation decisions

Accept

Reviewer #3: I would like to express my sincere gratitude for the opportunity to review and learn from the work presented, which is of utmost scientific relevance.

In this regard, I would like to raise some questions that I believe could contribute to the improvement of the study methodology.

Firstly, the authors may consider updating their search strategy, as it appears that the searches were only conducted until June 2022. Secondly, I noticed that the exclusion of 29 studies was based on pathway models, despite the fact that PROSPERO allowed for their inclusion (https://www.crd.york.ac.uk/PROSPERO/display_record.php?RecordID=199875). It would be helpful if the authors could clarify the reason for this discrepancy.

Additionally, I would like to bring to the authors' attention that the methodology of a systematic review does not permit exclusion of studies based on language or geographic location without adequate justification. Thus, I suggest that the authors include studies in English, Portuguese, and Spanish in their analysis.

Furthermore, I would appreciate a justification for the use of the National Institute for Health and Care Excellence (NICE) tool, which was replaced by the JBI tool. Also, I am curious to know if the authors searched EMBASE with MESH terms, as this appears to be an incorrect approach.

Moreover, could the authors kindly specify which tool or software was used for study extraction and management? Additionally, I found the definition of "disease-level constrained maximization of health decision rule" to be ambiguous, and I suggest that the authors clarify it using more straightforward language.

Lastly, I would strongly recommend that the authors evaluate the quality of evidence in their systematic review using the GRADE tool.

7. PLOS authors have the option to publish the peer review history of their article (what does this mean?). If published, this will include your full peer review and any attached files.

Reviewer #2: **Yes: **Mohnad Abdalla

Reviewer #3: **Yes: **Nathalia Sernizon Guimarães

---

## [Author Response · Author response to Decision Letter 1]

24 Jul 2023

Please see attached response letter.

---

## [Decision Letter · Decision Letter 2]

29 Aug 2023

A systematic review of whole disease models for informing healthcare resource allocation decisions

PONE-D-22-22167R2

Dear Dr Jin,

We’re pleased to inform you that your manuscript has been judged scientifically suitable for publication and will be formally accepted for publication once it meets all outstanding technical requirements.

Kind regards,

Karina Cardoso Meira, Ph.D

Academic Editor

PLOS ONE

Additional Editor Comments (optional):

Congratulations to the authors for the work of great relevance and excellence. After careful review your manuscript was considered for publication in the journal Plos one

Reviewers' comments:

Reviewer's Responses to Questions

**Comments to the Author**

1. If the authors have adequately addressed your comments raised in a previous round of review and you feel that this manuscript is now acceptable for publication, you may indicate that here to bypass the “Comments to the Author” section, enter your conflict of interest statement in the “Confidential to Editor” section, and submit your "Accept" recommendation.

Reviewer #2: All comments have been addressed

Reviewer #3: All comments have been addressed

2. Is the manuscript technically sound, and do the data support the conclusions?

Reviewer #2: Yes

Reviewer #3: Yes

3. Has the statistical analysis been performed appropriately and rigorously? 

Reviewer #2: Yes

Reviewer #3: Yes

4. Have the authors made all data underlying the findings in their manuscript fully available?

Reviewer #2: Yes

Reviewer #3: Yes

5. Is the manuscript presented in an intelligible fashion and written in standard English?

Reviewer #2: Yes

Reviewer #3: Yes

6. Review Comments to the Author

Reviewer #2: Please use the space provided to explain your answers to the questions above. You may also include additional comments for the author, including concerns about dual publication, research ethics, or publication ethics. (Please upload your review as an attachment if it exceeds 20,000 characters) (Limit 100 to 20000 Characters) No Comment

Reviewer #3: I believe that all the points have been carefully answered. I would like to congratulate you on your persistence.

7. PLOS authors have the option to publish the peer review history of their article (what does this mean?). If published, this will include your full peer review and any attached files.

Reviewer #2: **Yes: **Mohnad Abdalla

Reviewer #3: **Yes: **Nathalia Sernizon Guimarães

---

## [Editor Report · Acceptance letter]

31 Aug 2023

PONE-D-22-22167R2 

A systematic review of whole disease models for informing healthcare resource allocation decisions 

Dear Dr. Jin:

I'm pleased to inform you that your manuscript has been deemed suitable for publication in PLOS ONE. Congratulations! Your manuscript is now with our production department. 

Kind regards, 

on behalf of

Dr. Karina Cardoso Meira 

Academic Editor

PLOS ONE